

# Response of a semi-enclosed sea to perturbed freshwater and open ocean salinity forcing

Lars Arneborg[1], Magnus Hieronymus[1], Per Pemberton[1], Ye Liu[1], and Sam T. Fredriksson[1]

[1]Department of Research and Development, Swedish Meteorological and Hydrological Institute, Norrköping, 601 76, Sweden

*Correspondence to*: Lars Arneborg (lars.arneborg@smhi.se)

**Abstract.** The sensitivity of Baltic Sea salinities to changed fresh water forcing and other forcing factors have been debated during the last decades, since changed salinities would have large impacts on the marine ecosystems, and since this parameter still shows a high degree of uncertainty in regional climate projections. In this study we performed a sensitivity study where fresh water forcing and salinities at the outer boundaries of the North Sea were perturbed in a systematic way in order to obtain

a second-order Taylor polynomial of the statistical steady state mean salinity. The polynomial was constructed based on perturbations of a 57-year long hindcast run for the period 1961-2017 with a regional ocean model covering the North Sea and the Baltic Sea. The results show that the Baltic sea is highly sensitive to fresh water forcing and that only about one third of the boundary salinity change propagates into the Baltic Sea. The results are also analysed in terms of a total exchange flow analysis in the entrance region, and it is found that the Baltic Sea salinity sensitivity to a large degree can be explained by

increased freshwater input causing (1) dilution inside the Baltic Sea, (2) decreased inflows caused by changes to the mean sea level gradient in the entrance region, and (3) reduced inflow salinities due to recirculation of outflowing Baltic water in the entrance region where the inflow water consists of about two parts outflowing Baltic water and one part North Sea water.

## 1 Introduction

The Baltic Sea, Fig. 1, is a brackish, semi-enclosed sea in northern Europe, connected to the North Sea through a narrow and

shallow entrance area. Marine ecosystems in the Baltic sea have adjusted to the brackish (< 12 psu) conditions through thousands of years, and salinity changes are expected to have large impacts on ecosystems. For example, a cumulative impact study showed that a projected decrease in salinity together with increased temperatures and decreasing sea ice cover are as important as the cumulative impacts of all other present anthropogenic pressures (Wåhlström et al. 2022). Projections of future salinities show a large spread, where most regional ocean downscalings show a salinity decrease due to increased future

precipitation, whereas some of those that also include sea level rise show increased salinities due to larger inflows of salt (e.g. Meier et al. 2022). In order to reduce the spread there is a need of better understanding of the relative importance of various processes that influence the Baltic Sea salinity.




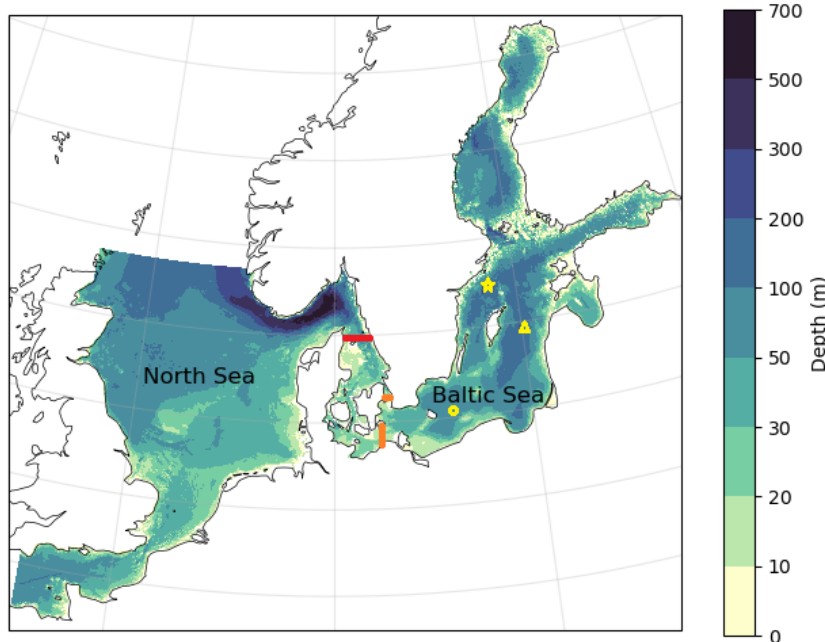


**Figure 1: Model bathymetry for the North Sea/Baltic Sea region with red line indicating position of the Northern Kattegat (NK) transect and orange line indicating the position of the sill transects. Yellow markers show the position of observation stations BY5 (circle), BY15 (triangle), and BY31 (star).**

Salinities in the Baltic Sea are the result of a balance between precipitation and evaporation over the Baltic sea and its watershed, inflow of saline water through the entrance, and outflow of mixed water (e.g Lehmann et al. 2022). This leads to a predominatly salinity stratified water body, although seasonally a thermocline develops in the mixed layer over most parts of the Baltic Sea. Mean fresh water net input is about 15000-16000 $m^3$/s (Meier et al. 2019). The inflow of saline water from the Kattegat is of about the same magnitude (Knudsen 1900, Burchard et al. 2018).  It is mainly barotropic, i.e. caused by wind-

forced sea level differences between southern Kattegat and the south-western Baltic Sea (Mohrholz 2018). Smaller barotropic inflows occur regularly, and mainly through the Sound, and do explain the main part of the salt import to the Baltic sea, whereas larger inflows, so-called Major Baltic Inflows (BMIs) that are dense enough to modify the Baltic deepwater, flow through both the Sound and the Belts, and do occur much less frequently and mainly during winter time (Mohrholz 2018).

Analysis of historical records and model results indicate a large sensitivity of Baltic Sea salinities to fresh water input. Rodhe

and Winsor (2003) propose a more than 3% salinity decrease for a 1% fresh water increase based on observational data.



Modeling studies show somewhat smaller sensitivities, 1 - 1.5% (Stigebrandt and Gustafsson 2003, Meier and Kauker 2003) salinity decrease for a 1% fresh water increase.

Based on a long hindcast simulation, Radtke et al. (2020) claim that multidecadal surface salinity variability in the Baltic sea is mainly caused by modulated precipitation over the watershed, with about 27% being caused by direct dilution of the Baltic

sea and the rest being caused by reduced inflows of salt. Meier et al. (2023) find that the reduced inflows of salt are dominated by reduced inflow salinities due to increased freshwater volumes in Kattegat, which is in contrast to the model study of Stigebrandt and Gustafsson (2003) where the main salt inflow decrease is assumed to be caused by decreased inflow volumes. In this study we focus on the sensitivity of Baltic Sea salinities to net freshwater input to the Baltic Sea and North Sea, and to variability of salinities at the boundary of the North Sea. We use a new 3D model setup with improved representation of Baltic

Sea inflows, and define the sensitivity experiment in order to estimate a second order Taylor polynomial based on the two variables fresh water input and boundary salinity. We also use total exchange flow analysis (e.g. Burchard et al. 2018) to quantify the physical processes that cause the sensitivity. One difference of this sensitivity experiment to previous model experiments is the inclusion of the North Sea, where other models have had a boundary in eastern Skagerrak/ northern Kattegat, and that we analyse the total exchange flows in two transects in the entrance regions to estimate water modification in the

Kattegat and Belt Sea region.

## 2 Methods

### 2.1 Experimental design: Taylor expansion

As mentioned in the introduction we are interested in the response of the Baltic Sea salinity to perturbations in precipitation/runoff and boundary salinity. Moreover, in this section we restrict our attention to study this response in a

statistical steady state.

Let $\bar{S}$ denote a long-time averaged salinity property, for example the basin mean salinity, defined through

$$\bar{S} = \frac{1}{V(t_2 - t_1)} \int_{t_1}^{t_2} dt \int_V s\, dV, \tag{1}$$

where $V$ is the volume of the basin, $t_1$ and $t_2$ the start and end times of the averaging period and $s$ the salinity. To study the response of $\bar{S}$ to the aforementioned perturbations we approximate $\bar{S}$ using a second order Taylor polynomial

$$\bar{S} \approx \bar{S}(SB_0, RP_0) + \frac{\partial \bar{s}}{\partial SB}(SB - SB_0) + \frac{\partial \bar{s}}{\partial RP}(RP - RP_0) + \frac{1}{2}\frac{\partial^2 \bar{s}}{\partial SB^2}(SB - SB_0)^2 + \frac{1}{2}\frac{\partial^2 \bar{s}}{\partial RP^2}(RP - RP_0)^2 + \frac{\partial^2 \bar{s}}{\partial SB \partial RP}(SB - SB_0)(RP - RP_0), \tag{2}$$

where $SB$ and $RP$ denote the boundary salinity and runoff+precipitation relative to the unperturbed state respectively, and the 0 subscript denotes the unperturbed state (i.e. long-time averages from a historical simulation).



The choice of a second order Taylor polynomial as opposed to a simpler first order one is motivated by the fact that it gives us
a chance to quantify the interaction between the two forcing terms through the cross-derivative term. To compute the coefficients for the Taylor polynomial, six runs are needed in addition to the unperturbed hindcast giving the $\bar{S}(SB_0, RP_0)$ term. The scheme is as follows:

$$\frac{\partial \bar{S}}{\partial SB} = (2h)^{-1}(\bar{S}(SB_0 + h, RP_0) - \bar{S}(SB_0 - h, RP_0)), \tag{3}$$

$$\frac{\partial \bar{S}}{\partial RP} = (2k)^{-1}(\bar{S}(SB_0, RP_0 + k) - \bar{S}(SB_0, RP_0 - k)), \tag{4}$$

$$\frac{\partial^2 \bar{S}}{\partial SB^2} = h^{-2}(\bar{S}(SB_0 + h, RP_0) - 2\bar{S}(SB_0, RP_0) + \bar{S}(SB_0 - h, RP_0)), \tag{5}$$

$$\frac{\partial^2 \bar{S}}{\partial RP^2} = k^{-2}(\bar{S}(SB_0, RP_0 + k) - 2\bar{S}(SB_0, RP_0) + \bar{S}(SB_0, RP_0 - k)), \tag{6}$$

$$\frac{\partial^2 \bar{S}}{\partial SB \partial RP} = (2hk)^{-1}(\bar{S}(SB_0 + h, RP_0 + k) - \bar{S}(SB_0 + h, RP_0) - \bar{S}(SB_0, RP_0 + k) + 2\bar{S}(SB_0, RP_0) - \bar{S}(SB_0 - h, RP_0) -$$
$$\bar{S}(SB_0, RP_0 - k) + \bar{S}(SB_0 - h, RP_0 - k)), \tag{7}$$

where $h$ is a salinity perturbation on the outer boundary, and $k$ is a fractional perturbation of the precipitation and runoff. This
means that in addition to the reference case $\bar{S}(SB_0, RP_0)$, the following integrations are needed: $\bar{S}(SB_0+h, RP_0)$, $\bar{S}(SB_0-h, RP_0)$, $\bar{S}(SB_0, RP_0+k)$, $\bar{S}(SB_0, RP_0-k)$, $\bar{S}(SB_0+h, RP_0+k)$ and $\bar{S}(SB_0-h, RP_0-k)$. The perturbations $h$ and $k$ are introduced as follows: $h$ is a 0.5 psu change to the boundary salinity and $k$ is a fractional change of 0.2 applied to precipitation and runoff. $RP+k$ runs thus have their runoff and precipitation multiplied by 1.2 and $RP-k$ ones by 0.8. The $SB+h$ and $SB-h$ runs correspond to increases and decreases in boundary salinity by 0.5 psu, respectively. Note also that while $\bar{S}$ is used as an example variable here, one
can calculate Taylor polynomials for any variable of interest using the same runs. A further note on the computation is that we use runs over the period 1961-2017, with $t_1 = 1990$ and $t_2 = 2017$, which gives a thirty-year long spin-up, a period similar to the estimated flushing time scale of the basin (e.g Meier and Kauker 2003). All experiments are started from the same initial state, which is a 30-year spinup from climatology T/S and ocean at rest.

To test the interpolation qualities of the polynomial out of sample we also integrated three runs $\bar{S}(SB_0+2h, RP_0+2.5k)$,
$\bar{S}(SB_0+1.2h, RP_0+0.5k)$, and $\bar{S}(SB_0-1.2h, RP_0+0.5k)$, which are not used to calculate any coefficient in the Taylor polynomial and are thus independent, measures of the ability of the extrapolation/interpolation method.

Note that a relative change in precipitation and runoff is not the same as a relative change in net freshwater input to a system, $Q_f$, since a large fraction of the precipitation is lost to evaporation, making $Q_f/Q_{f0}$ smaller than $RP$. Evaporation is relatively constant between the different runs in this experiment, so the change in net freshwater input is about equal to the change in
precipitation and runoff, but since $Q_f$ is smaller than precipitation + runoff, the relative change in $Q_f$ is about 28% when the change in $RP$ is 20%.



## 2.2 Total exchange flow analysis

Analysing transports through transect in salinity coordinates rather than Eulerian vertical coordinates (e.g. Walin 1977, MacCready 2011, Burchard et al. 2018) enables quantification of water mass transformation and estuarine water exchange in a way very similar to the ideas behind the Knudsen theorem (Knudsen 1900). The time averaged advective inflows of volume and salt mass in water with salinity larger than $s$ can be written as

$$Q(s) = \overline{\int_{A(s)} u \, dA}, \tag{8}$$

and

$$F(s) = \overline{\int_{A(s)} s u \, dA}, \tag{9}$$

where $A(s)$ at any time is the transect area with salinity larger than $s$, $u$ is the horizontal velocity component normal to the transect pointing into the estuary, and overbar denotes time averaging. These can be integrated into in- and outflows through

$$Q_{in} = \int_0^\infty max(0, -\tfrac{\partial Q}{\partial s}) ds, \tag{10}$$

$$Q_{out} = \int_0^\infty max(0, \tfrac{\partial Q}{\partial s}) ds, \tag{11}$$

$$F_{in} = \int_0^\infty max(0, -\tfrac{\partial F}{\partial s}) ds, \tag{12}$$

$$F_{out} = \int_0^\infty max(0, \tfrac{\partial F}{\partial s}) ds, \tag{13}$$

where

$$max(x,y) = \begin{cases} x & , x \geq y \\ y & , x < y \end{cases}. \tag{14}$$

Now, mean inflow and outflow salinities can be calculated as

$$S_{in} = \frac{F_{in}}{Q_{in}}, S_{out} = \frac{F_{out}}{Q_{out}}. \tag{15}$$

When averaged over long time relative to the residence times of salt and freshwater in the system, and changes to inflow and fresh-water input, the accumulation of volume and salt are small relative to the in- and outflow transports, and a steady state volume and salt budget analysis results in the Knudsen relations (see Burchard et al. 2018 for a discussion)

$$Q_{in} = \frac{S_{out}}{S_{in} - S_{out}} Q_f, \tag{16}$$

$$Q_{out} = \frac{S_{in}}{S_{in} - S_{out}} Q_f, \tag{17}$$



where $Q_f$ is the mean integrated fresh-water input (runoff + precipitation – evaporation) to the estuary inside the investigated transect.

For two transects, A and B, where B is located closest to the open ocean, part of the outflow through section $A$, $Q_{RA}$, is recirculated and contribute to the inflow through section $A$, while the rest, $Q_{outA} - Q_{RA}$, flows out through section $B$. Similarly, part of the inflow through section $B$, $Q_{RB}$, is recirculated and contribute to the outflow through section $B$, while the rest, $Q_{inB} - Q_{RB}$, contributes to the inflow though section $A$. The corresponding salt and volume inflows through section $A$ can be written as

$$Q_{inA}S_{inA} = (Q_{inB} - Q_{RB})S_{inB} + Q_{RA}S_{outA}, \tag{18}$$

$$Q_{inA} = Q_{inB} - Q_{RB} + Q_{RA} + \gamma\Delta Q_f, \tag{19}$$

where $\Delta Q_f$ is the fresh water input between section $A$ and $B$ and $\gamma$ is the fraction of this that enters the Baltic Sea. The recirculation fluxes can now be found as

$$Q_{RA} = \frac{Q_{inA}(S_{inB} - S_{inA}) - \gamma\Delta Q_f S_{inB}}{S_{inB} - S_{outA}}, \tag{20}$$

$$Q_{RB} = \frac{Q_{outB}(S_{outB} - S_{outA}) + (1-\gamma)\Delta Q_f S_{outA}}{S_{inB} - S_{outA}}. \tag{21}$$

These expressions are similar to the efflux/reflux expressions first presented by Cokelet and Stewart (1985), except for the addition of local runoff. They basically express that the outflows from the region bounded by the A and B sections are a result of turbulent mixing of the inflowing water masses to the region. In- and outflows, and recirculation fluxes, were calculated as described above for two transects, Fig. 1, one in northern Kattegat (section $B$), and one that includes the Darss- and Drogden sills at the rim of the Baltic Sea (section $A$). The recirculation fluxes are therefore caused by water mass modifications in Kattegat or in the Belts and the Sound. Temporal averages were based on the period 1990 - 2017.

## 2.3 Model setup

Our experiments use a NEMO 4.2 configuration of the North Sea and Baltic Sea region (see Fig. 1). The configuration builds on a previous NEMO 3.6 configuration with evaluations of ocean and sea-ice parameters presented in Hordoir et al. (2019) and Pemberton et al. (2017), respectively. The horizontal resolution is $0.055°$ in the zonal and $0.033°$ in the meridional direction, which amounts to a nominal resolution of 3.7 km (2 nautical miles). To adapt the configuration to NEMO 4.2 we have changed the tracer advection scheme to 4th order Flux Corrected Transport (FCT) scheme, and the lateral viscosity formulation from a laplacian to a bi-laplacian operator. To improve the salinity dynamics of the model configuration we have changed the vertical coordinate system from the geopotential Z-partial steps formulation (used in Hordoir et al. 2019) to the terrain-following Multi-Envelope s-coordinate (MEs) system developed by Brucaferri et al. (2018). The MEs coordinates are configured to use 2 envelopes. The upper envelope ranges between the surface and 250 m and uses 43 levels, and the lower envelope all depths





below 250 m using 13 levels. We thus retain the 56 vertical levels used in Hordoir et al. (2019) but gain a higher vertical
resolution in the Baltic Sea. In a separate study (Pemberton et al, manuscript in preparation for GMD) we make a thorough
evaluation of the impact of the vertical coordinate system on the Baltic Sea hydrography and circulation. Here we instead
restrict our evaluation to compare the mean vertical salinity distribution and surface/bottom salinity evolution for the hindcast
run to observations at three stations.

The hindcast run is the reference simulation for the perturbation runs described in Section 2.4. It is a run from 1961 to 2017
based on the best available forcing. Before performing the hindcast and perturbation runs, we spun up the model circulation
by using initial conditions for salinity and temperature from the Janssen et al. (1999) climatology, with ocean currents and sea
surface height set to zero. Then we ran the model three cycles repeating the same 10 years period using atmospheric, runoff
and open boundary forcing from 1961–1970 so that model dynamics reached near-equilibrium level. The meteorological
forcing was derived from the UERRA regional reanalysis (Dahlgren et al., 2016), which offers a spatial resolution of 11 km
and a temporal resolution of 1 hour for parameters such as wind, air pressure, air temperature, humidity, and both solar and
long-wave downward radiation. Precipitation (rain and snow) is provided at a temporal resolution of 12 hours. River runoff
forcing was provided as daily values from a dedicated simulation with the Hydrological Predictions for the Environment model
with the European application v.3.1.8 (E-HYPE; Donnelly et al., 2016). The open boundary forcing includes barotropic
currents, sea level, nine tidal constituents, and monthly salinity and temperature data. The barotropic currents and sea level
were calculated using the 2D North Atlantic Model (NOAMOD; She et al., 2007) storm-surge model. These values were
adjusted for baroclinic influences by incorporating monthly sea level data from the European Centre for Medium-Range
Weather Forecasts' Ocean Reanalysis System 4 (ORAS4), which enhances the description of the North Sea ocean circulation.
The salinity and temperature profiles are monthly mean values interpolated from the ORAS4 configuration (Balmaseda et al.,
2013).
To illustrate the model performance, observations were taken from the Swedish archive for oceanographic data (SHARK,
www.smhi.se), at three selected stations in the Baltic Sea, located at different distances from the entrance (see Fig. 1). Vertical
salinity profiles from the three stations (Fig. 2) show that the salinity stratification and the range of variability are well described
by the model at the three stations. The temporal variability in the surface water and in the bottom water (Fig. 3) show that
surface water salinity is well represented by the model both with respect to short-term variability and long-term trends. Both
model and observations show a decrease in surface water salinity between 1980 and 2000. Before 1980 the model surface
salinities are slightly larger than the observed salinities, which may be due to an adjustment from too high initial salinities.
Near-bottom water model salinities also show similar temporal variability as in observations with a long period between 1980
and 1994 without strong inflows of saline water.



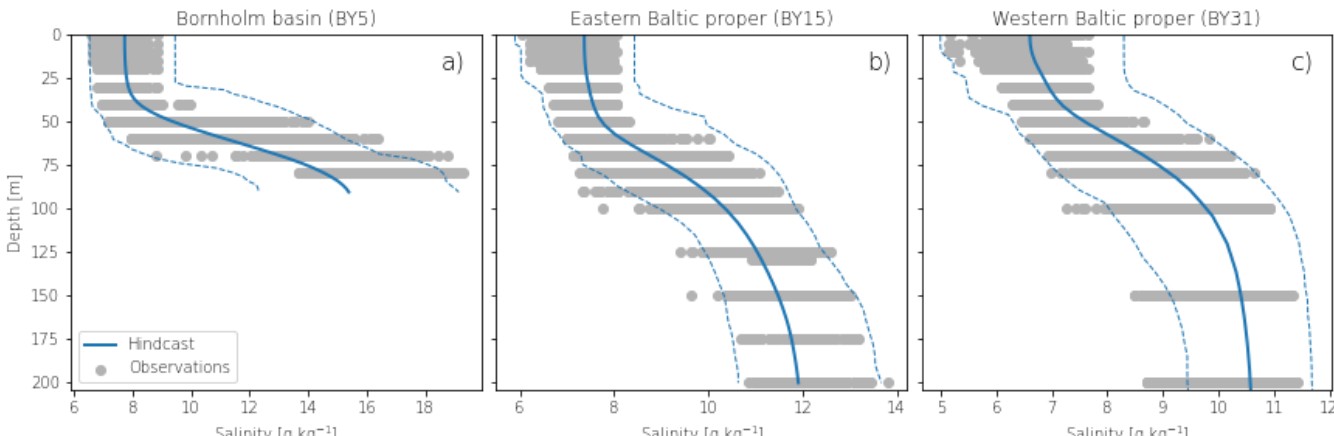

**Figure 2: Salinity profiles at three selected stations in the Baltic Sea. The blue solid lines represent the temporal mean of the hindcast experiment, and the dashed lines the 5th and 95th percentiles. The grey dots represent observed salinity from the SHARK database.**

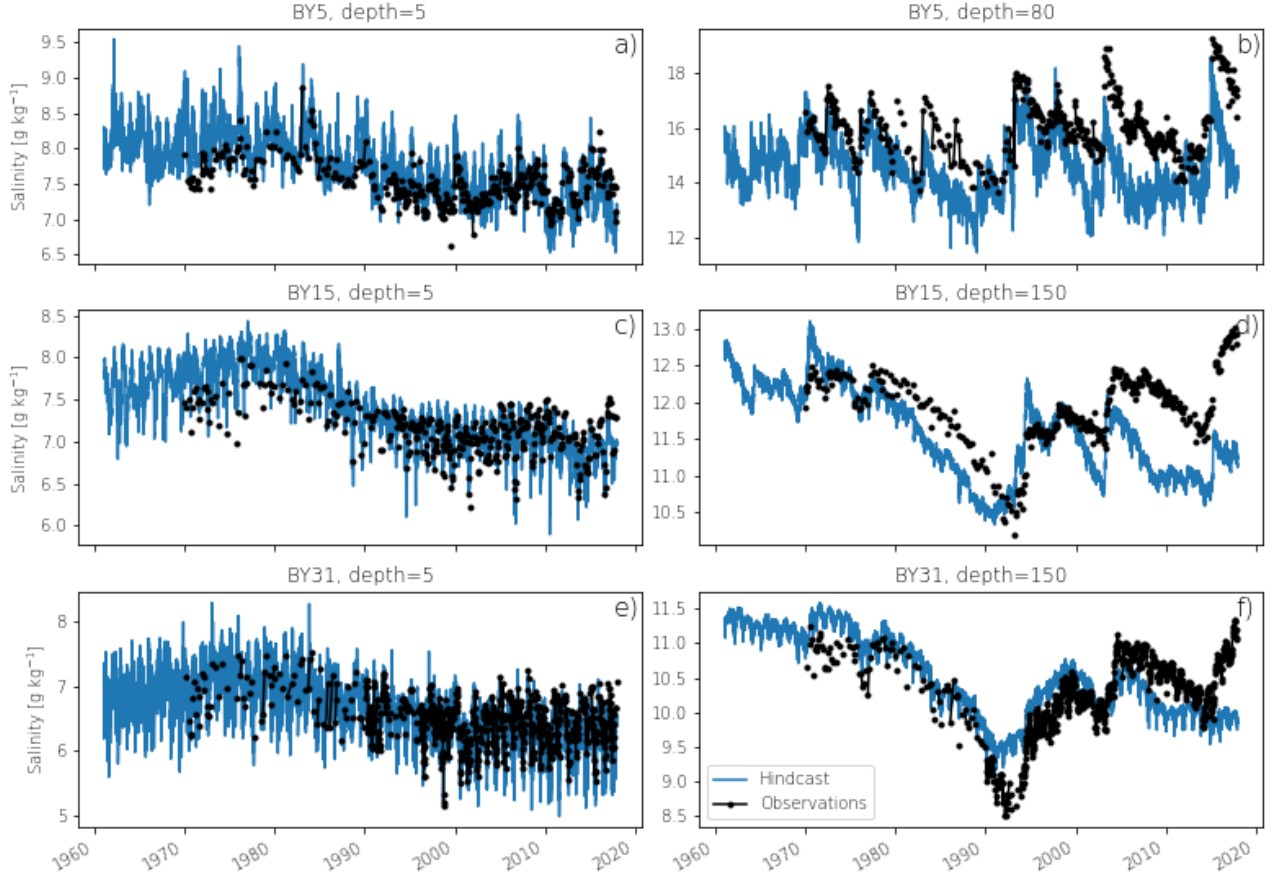

**Figure 3:  Time-evolution of surface water (left column) salinity and bottom water (right column) salinity at three selected stations in the Baltic Sea. The blue solid lines are from the hindcast experiment and the black dots observations from the SHARK database.**




## 2.4 Model runs

To investigate the salinity dynamics of the Baltic Sea in response to variations in freshwater input and boundary salinity using a Taylor expansion approach (Sect. 2.1), we performed a series of model simulations in which these two factors were systematically perturbed over the period from 1961 to 2017, see Table 1. In these runs, either boundary salinities were
maintained the same as in the reference run while freshwater input was varied by ±20% relative to the reference scenario (runs RP+ and RP-), or boundary salinities were altered by ±0.5 psu compared to the reference experiment while runoff and precipitation was maintained equal to those of the reference run (runs SB+ and SB-). The fractional changes in freshwater input were applied using NEMO namelist options for runoff and precipitation multipliers. To capture interaction terms, two experiments were performed (RP+SB+ and RP-SB-). Finally, three additional runs were carried out to evaluate the
polynomial's validity outside the bounds of the runs used to determine the polynomial. These involved; a 50% increase in freshwater input and a 1.0 psu rise in boundary salinity from the reference run (RP++SB++), a 10% increase in freshwater input and 0.6 psu rise (RP+10SB+06) and 0.6 psu decrease (RP+10SB-06) in boundary salinity.

**Table 1: Model runs performed**

| Run | Boundary Salinity | Runoff + Precipitation |
|---|---|---|
| 0: CTL | Hindcast | Hindcast |
| 1: RP+ | Hindcast | Hindcast*1.2 |
| 2: RP- | Hindcast | Hindcast*0.8 |
| 3: SB+ | Hindcast + 0.5 | Hindcast |
| 4: SB- | Hindcast - 0.5 | Hindcast |
| 5: RP+SB+ | Hindcast + 0.5 | Hindcast*1.2 |
| 6: RP-SB- | Hindcast - 0.5 | Hindcast*0.8 |
| 7: RP++SB++ | Hindcast + 1.0 | Hindcast*1.5 |
| 8: RP+10SB+06 | Hindcast + 0.6 | Hindcast*1.1 |
| 9: RP+10SB-06 | Hindcast – 0.6 | Hindcast*1.1 |





# 3 Results

## 3.1 Temporal and spatial salinity changes

The time series of yearly mean salinities averaged over the whole Baltic Sea volume inside the sill transect (Fig. 1) are shown in Fig. 4. It is clearly seen that the salinities show a large sensitivity towards fresh water forcing, with a difference from the hindcast run of about 2 psu. The changes caused by changing boundary salinity are much smaller.

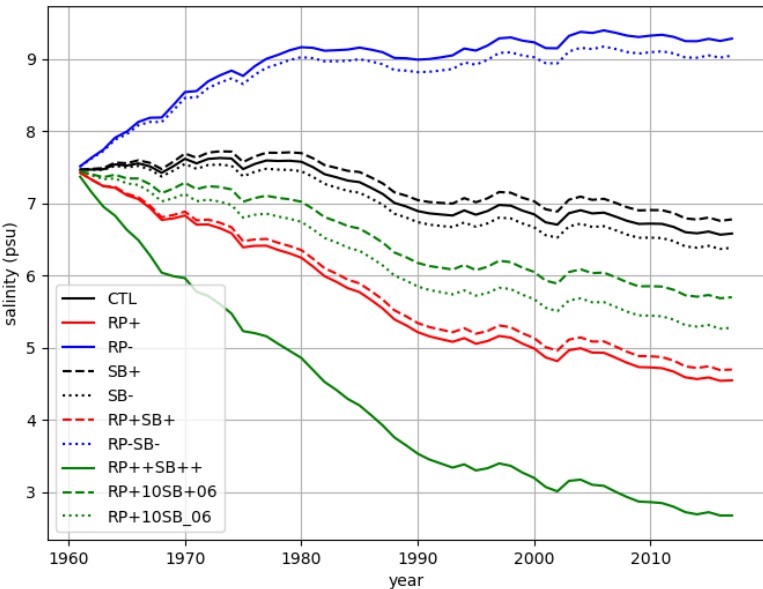


**Figure 4: Yearly mean salinities for the Baltic sea averaged over the volume inside the sill transect.**

Figure 5 shows maps of depth averaged salinity changes relative to the hindcast run, averaged over the period 1990-2017. For the precipitation and evaporation change (RP+ and RP-) runs, the salinity changes are larger inside the Baltic Sea than outside and rather homogeneous, however gradually somewhat decreasing as we move north. The changes for these runs are almost equal but of opposite sign, except for the Gulf of Bothnia where the salinity decrease is smaller for increasing runoff than the salinity increase for decreasing runoff. In Kattegat, changes are mainly seen in the shallow areas towards the Danish coast and close to the Swedish coast, and in Skagerrak and the North Sea, changes are mainly concentrated in the coastal areas.

The boundary salinity changes (SB+ and SB-) manifest themselves rather uniformly in the North Sea, Skagerrak and Kattegat. Inside the Baltic Sea, the changes are much smaller but still distributed rather uniformly.





For combinations of the perturbations, the changes caused by runoff and precipitation dominate in the Baltic Sea, Kattegat and along the coasts of Skagerrak and the North Sea, while the boundary salinity changes dominate in the open parts of the North Sea and Skagerrak.


**Figure 5: Changes relative to hindcast run for depth averaged mean salinities for the period 1990-2017.**





## 3.2 Taylor expansion

Figure 6, shows the second order Taylor polynomial for $\bar{S}$. The coefficients of the polynomial are given in Table 2. It is quite clear from the figure that the response to perturbations is linear in $h$ (boundary salinity change), and nonlinear in $k$ (runoff and

freshwater input change). The latter is seen through the increasing distance between level curves as $k$ increases. This conclusion is further supported by comparing the ratio of the magnitude of the first to the second derivatives of $\bar{S}$ with respect to SB and RP, which is 191 for the SB case and 0.39 for the RP case (Table 2). The quadratic term is thus much more important in the RP case. The interaction term, the cross derivative is also relatively important. Putting it to zero gives a polynomial (not shown) with deviations from the full polynomial notable especially for large values of $k$. It is also clear from Fig. 4 that boundary

salinity changes are damped in the basin. The first derivative of $\bar{S}$ with respect to SB is 0.36, indicating that a salinity change of 1 psu at the boundary gives a change in $\bar{S}$ of only 0.36 psu.

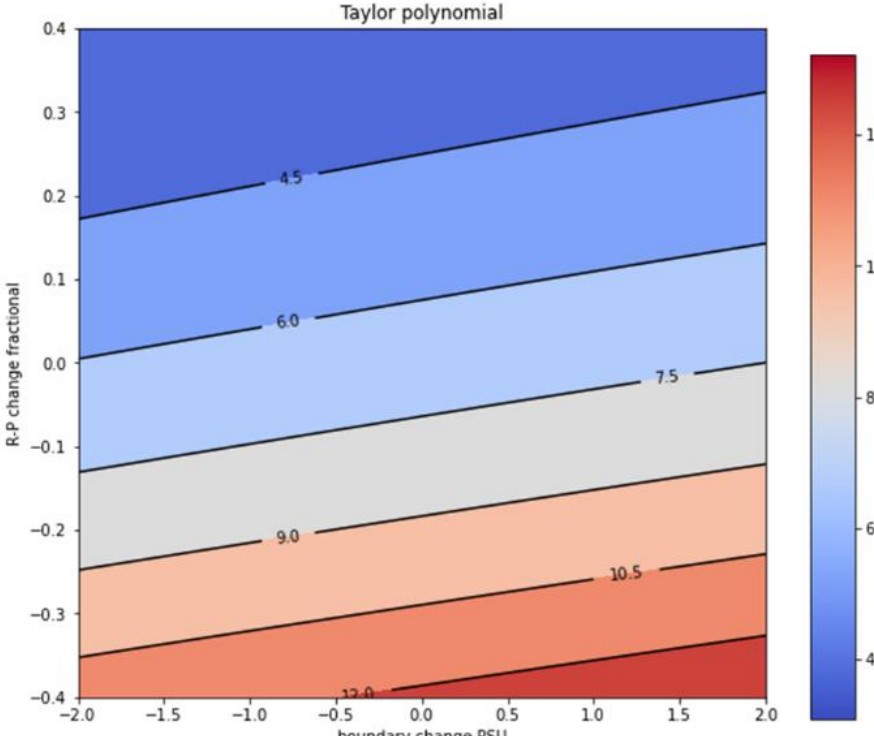

**Figure 6: Taylor polynomial for $\bar{S}$.**






**Table 2: Polynomial coefficients for second order polynomial of Baltic Sea mean salinities for the period 1990 – 2017.**

| Coefficient | Value | Unit |
|---|---|---|
| $\bar{S}$ | 6.78 | psu |
| $\dfrac{\partial \bar{S}}{\partial SB}$ | 0.36 | |
| $\dfrac{\partial \bar{S}}{\partial RP}$ | -10.86 | psu |
| $\dfrac{1}{2}\dfrac{\partial^2 \bar{S}}{\partial SB^2}$ | 0.00047 | psu$^{-1}$ |
| $\dfrac{1}{2}\dfrac{\partial^2 \bar{S}}{\partial RP^2}$ | 6.94 | psu |
| $\dfrac{\partial^2 \bar{S}}{\partial SB \partial RP}$ | -0.33 | |

The interpolation and extrapolation qualities of the polynomial are shown in Fig. 7. The scatter plot shows modeled $\bar{S}$ against

$\bar{S}$ derived from the Taylor polynomial. The point with the largest deviation is from to the $\bar{S}$ ($SB_0$+2$h$, $RP_0$+2.5$k$) run (RP++SB++) which is a large extrapolation. Even so the quality is rather good. Note also that only the point $\bar{S}(SB_0, RP_0)$ is exactly equal for the polynomial and the model, but almost all the other points also fall on a near perfect line, indicating very good interpolation properties. Looking beyond $\bar{S}$ one can also estimate Taylor polynomials for other quantities based on the present experiment, for example surface salinities, bottom salinities, or higher percentiles of the Baltic Sea salinity distribution.

As an example, the Taylor polynomial for the 95th percentile of the Baltic Sea spatial salinity distribution gives a larger response to boundary salinity change than the mean salinity, with first derivative of the 95% percentile with respect to $SB$ being 0.57 where it is 0.36 for $\bar{S}$. However, qualitatively, polynomials for different percentiles are relatively similar.





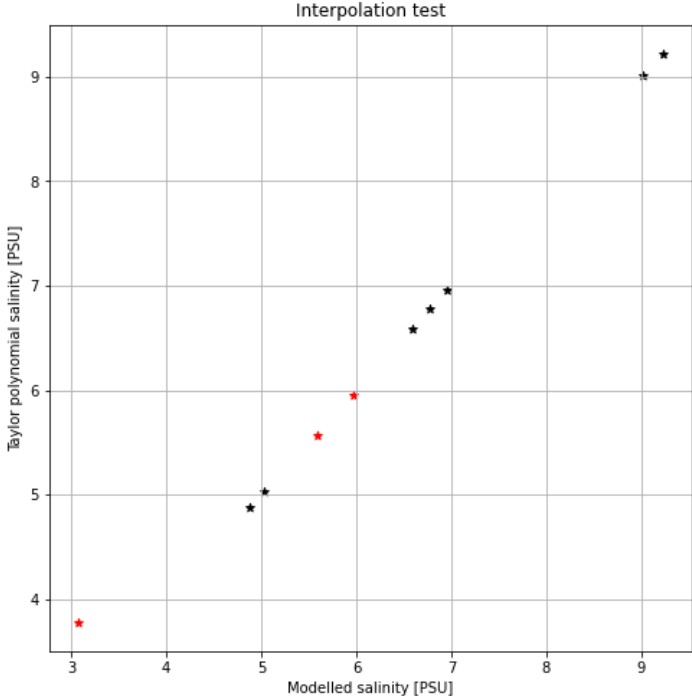

**Figure 7: Comparison of modeled and Taylor polynomial $\overline{S}$ at known points. Black stars are for runs used to calculate the polynomial**
**and red stars are for the remaining runs. The lowest salinity comes from the $\overline{S}$(SB+2h, RP+2k) run and is thus a rather large**
**extrapolation.**

## 3.3 Total exchange flow analysis

The inflow functions, $Q$ and $F$, for volume and salt, defined in (8) and (9), and averaged over the period 1990 to 2017, are
shown in Figs. 8 and 9.





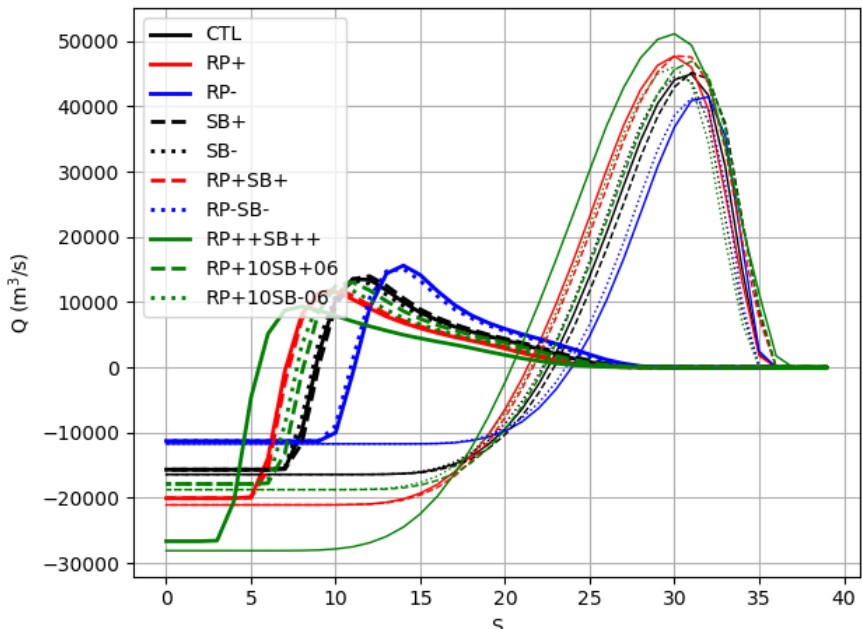

**Figure 8: Cumulated inflows of volume, $Q$, of water with salinity larger than $s$ as defined in (8). Thick lines are for transports through the sill transect, and thin lines are for transports through the northern Kattegat transect. Inflows happen at salinity intervals with negative gradient of $Q$, whereas outflows happen at salinity intervals with positive gradient.**

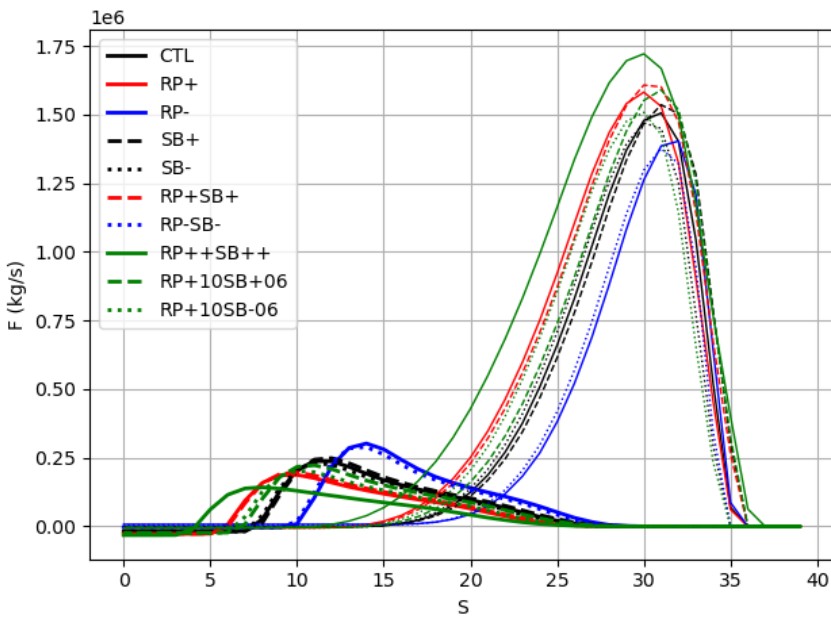


**Figure 9: Cumulated inflows of salt, $F$, in water with salinity larger than $s$ as defined in (9). Thick lines are for transports through the sill transect, and thin lines are for transports through the northern Kattegat transect. Inflows happen at salinity intervals with negative gradient of $F$, whereas outflows happen at salinity intervals with positive gradient.**



For the hindcast run, the net outflow, seen as $Q$ at low salinities, Fig. 8, is 15700 m$^3$/s at the sill transect and 16400 m$^3$/s at the northern Kattegat transect. The main outflows at the sill transect, seen as the part of $Q$ with positive gradient with respect to salinity, have salinities between 7 and 11 psu, and the main inflows have salinities between 11 and 28 psu. At the northern Kattegat transect, the main outflows have salinities between 15 and 31 psu while the inflows happen at 31 - 36 psu. The strength of the inflow, seen as the maximum value of $Q$, is about 13500 m$^3$/s at the sill transect and 45000 m$^3$/s at northern Kattegat.

For increased runoff and precipitation, RP+, RP+SB+, and RP++SB++, the inflow strengths through the sill transect are reduced and the salinities of inflows and outflows are also decreased. At the sill transect, the influence of fresh water input on inflows is seen to be larger for low saline inflows than for high-saline inflows. For the northern Kattegat transect, the inflow strength increases for these cases, the outflow salinities decrease, whereas the inflow salinities are less affected.

The salinity changes for cases with decreasing runoff and precipitation are opposite to those for the cases with increasing runoff and precipitation.

Changing boundary salinities are mainly affecting the salinities of inflows and outflows at the northern Kattegat boundary, but do cause less changes to the inflows and outflows at the sill transect, and only small changes to the inflow and outflow strengths. Figure 9 clearly illustrates that the overturning of salt in Kattegat and the Belts and Sound is much larger than within the Baltic Sea. Otherwise, it shows many of the same features as seen in Fig. 8. In a steady state condition, the net cumulated inflow of salt at small salinities should be zero. It is, however, seen that many of the curves have a small negative value there, i.e. a net outflow of salt, which corresponds to the negative mean salinity trend in the period 1990-2017 seen in Fig. 2.

The in- and out fluxes through the transects and the recirculation calculated from (20) and (21) are shown for the hindcast run in Fig. 10. The recirculation fluxes are calculated using $\gamma = 1$, i.e. assuming the local fresh water input between transects A and B is being added to the outflow through transect B. It is seen that the inflowing water to the Baltic Sea through transect A consists of about 64% outflowing Baltic Sea water and 36% inflowing Skagerrak water, and that the outflow through transect B consists of about 64% inflowing Skagerrak water, 1% local fresh water and 35% outflowing Baltic Sea water.



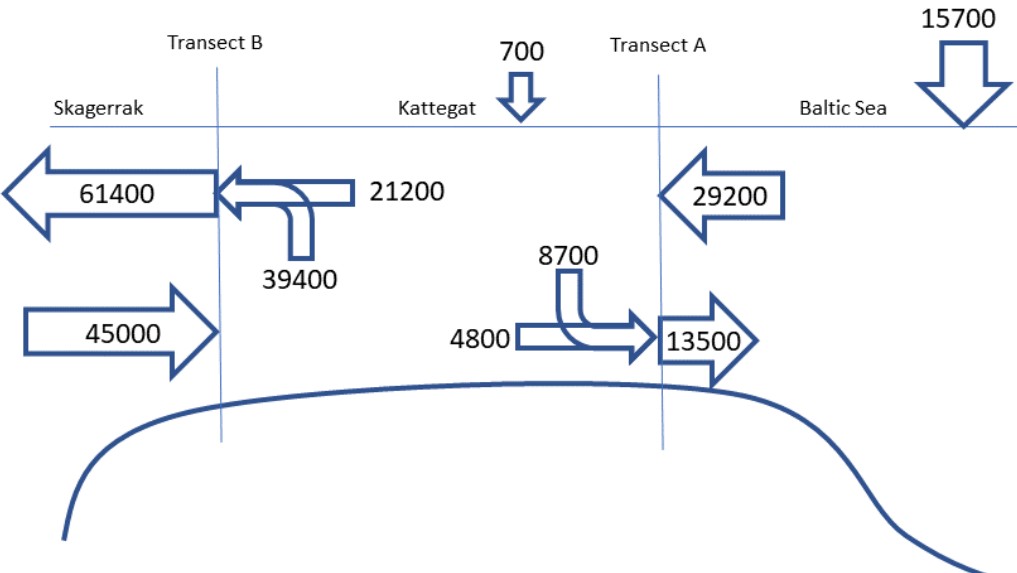

**Figure 10: Mean volume fluxes for the period 1990 - 2017 in m³/s. The vertical arrows show the net inputs of fresh water to the Baltic Sea inside transect A and to the region between transects A and B. Large horizontal arrows show in- and outflows through the transects. Smaller horizontal and bended arrows show the contributions from inflow (outflow) through transect B (A), and recirculated outflow (inflow) from transect A (B) to the inflow (outflow). The recirculation fluxes are calculated with $\gamma = 1$, in equations (20) and (21), i.e. assuming that the local freshwater contribution between transects A and B are added to the outflow through transect B.**

When modifying the runoff, precipitation and boundary salinity, these fluxes change. The changes are small for the SB+ and SB- experiments (not shown), but for the RP+ and RP- experiments the changes are larger. Figure 11 shows the changes to these fluxes for an increase in $Q_f$ of 100 m³/s, calculated from the central differences derivative of these fluxes with respect to $Q_f$ based on the RP+ and RP- experiments. It is seen that 54% of the increased net inflow of fresh water to the Baltic Sea cause increased outflows from the Baltic, while the remaining 46% cause decreased inflows to the Baltic Sea. The decreased inflow is caused by about 2/3 decreased recirculation and 1/3 decreased Skagerrak water. This means that the inflowing waters continue to consist of about 2/3 Baltic Sea water and 1/3 Skagerrak water also when the fresh water forcing changes. While the overturning of salt inside the Baltic Sea decreases due to decreasing inflows, the overturning within Kattegat increases, since both the inflows and outflows through the northern Kattegat transect increase. One way to explain this increased overturning may be the increased baroclinic pressure gradients between Skagerrak and Kattegat when the salinities in Kattegat decrease with increasing fresh water forcing.



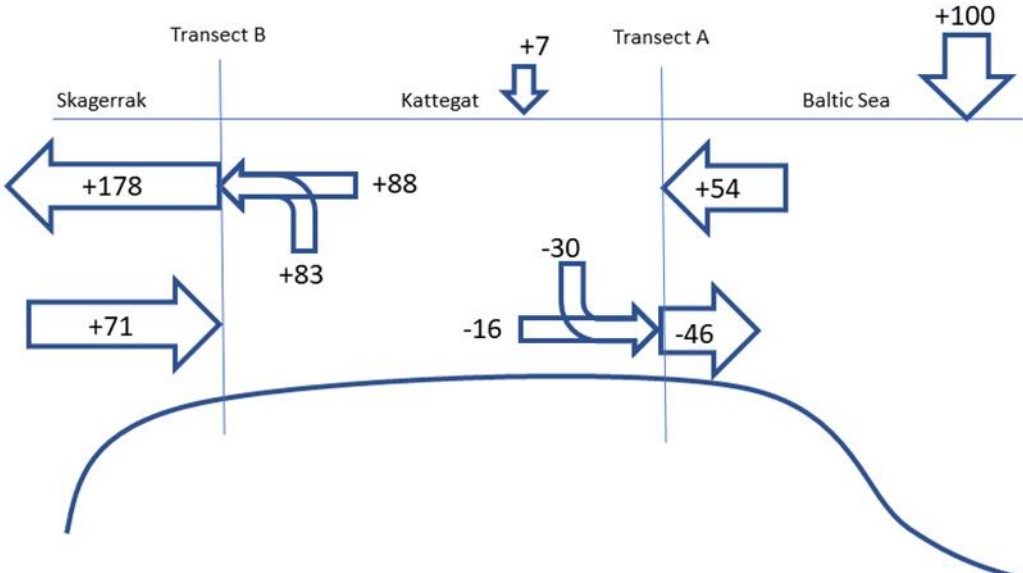

**Figure 11: Change in volume fluxes in m³/s for an increase in freshwater input to the Baltic Sea of $\Delta Q_f$ = 100 m³/s averaged over the period 1990-2017. The arrows are described in the Figure 8 caption.**

Figure 12 shows the probability density functions (pdf) for inflows and outflows through the sill transect for the hindcast and changed runoff and precipitation cases. It is seen that the changed freshwater basically shifts the pdf towards larger or smaller inflows while maintaining the same shape.

Since the curve is almost symmetric around zero, though shifted somewhat towards the negative side, shifting the curve laterally causes almost the same decrease or increase on the positive side as the increase or decrease on the negative side. This

explains why the changes in inflows and outflows (Fig. 11) are of about the same magnitude.

The dashed lines in Fig. 12 are pdfs obtained from the hindcast run time series with an addition or subtraction of a constant volume flux corresponding to the mean change in $Q_f$ for the RP+ and RP- cases. These are quite similar to the corresponding RP+ and RP- curves. This indicates that the changed in- and outflows are basically caused by a shift up or down of the time series with the mean change in $Q_f$.





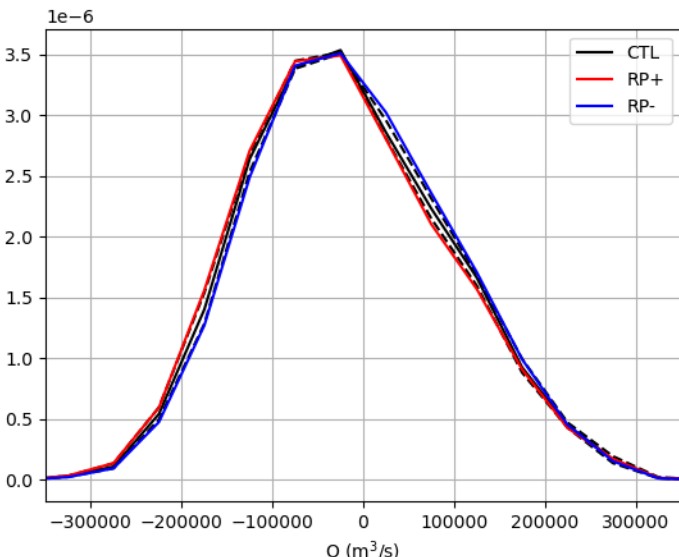


**Figure 12: Probability density function for inflows (net inflows at each time step) for the hindcast run and the RP+ and RP- runs. Dashed lines show the frequency distributions for the hindcast run with 4387 $m^3$/s added and subtracted from the hindcast run time series.**

One way to explain that an increased input of freshwater causes a more or less constant change to the fluctuating inflows is

that they cause a raised sea level relative to the reference situation, Fig. 13, which causes an extra outward barotropic flow. Similarly, a decreased input of freshwater will cause a lowered sea level relative to the reference situation and therefore smaller outward barotropic flows and increase inward barotropic flows.

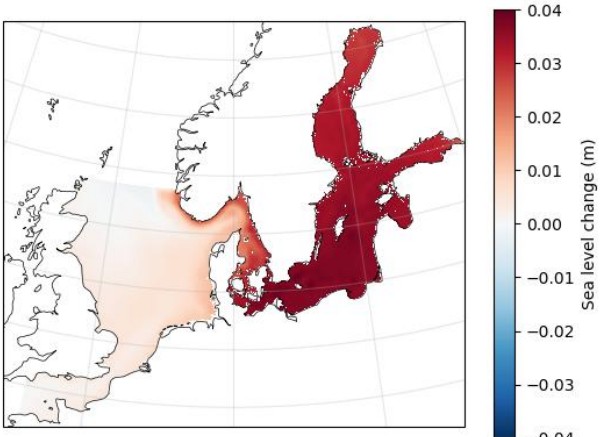

**Figure 13: Mean sea level difference (m) between run RP+ and CTL for the years 1990 – 2017.**



## 4 Summary and discussion

The sensitivity of the Baltic Sea steady state salinity was investigated with a series of 57 year runs with a regional ocean model for the Baltic and North Sea, consisting of a hindcast run, and perturbations of that with respect to boundary salinity and fresh water forcing.

A second order polynomial of the Baltic Sea mean steady state salinity in terms of the variables boundary salinity and fresh water forcing was constructed. Similar polynomials can be constructed for sensitivity of other parameters to these variables based on the present run. The results showed a large and non-linear sensitivity to precipitation and runoff, whereas the response to boundary salinity was linear. The response of the Baltic Sea to boundary salinity may be seen as small or large depending on perspective. Only 36% of the boundary salinity change was seen in the Baltic Sea, but since the salinity of the Baltic Sea is only about 22% of that in the North Sea, the relative change is actually larger in the Baltic Sea than in the North Sea.

Total exchange analysis was used to analyze inflows and outflows through two transects located over the sill and in the northern end of Kattegat, as well as the water mass transformation within Kattegat and the Belt sea. It was found that the inflowing water through the sill transect consisted of 36% Skagerrak water (inflow water through the northern Kattegat transect) and 64% outflowing Baltic Sea water. When modifying runoff and precipitation, this mixture remained relatively constant.

When increasing precipitation and runoff to the Baltic Sea, 54% of the increased net fresh water input were exported as increased outflows through the sill transect, whereas 46% resulted in decreased inflows. One way to explain this change is that the increased fresh water input causes a higher sea level in the Baltic Sea, and therefore a net change in sea level gradients over the inflow region which causes a change to the barotropic flows. With the large-amplitude fluctuations in in- and outflows taking place in the inflow region, such a more or less constant net change to the barotropic flows contributes almost equally to increased outflows and decreased inflows.

The above results show that the sensitivity of the Baltic Sea mean salinity to increased fresh water forcing is caused both by decreased inflow volume fluxes and by recirculation of the outflowing Baltic Sea water that causes fresher inflows. In a steady state, following Knudsen (1900), the net flow of salt is zero and the outflow of volume is equal to the net input of fresh water. To this can be added the result of the present work that the salinity of inflows through the sill transect can be written as

$$S_{in} = \alpha S_0 + (1-\alpha)S_{out}, \tag{22}$$

where $S_0$ is the salinity of the inflowing water from Skagerrak, $S_{out}$ is the salinity of outflowing water from the Baltic Sea through the sill transect, and $\alpha$ is the amount of Skagerrak water in the inflowing water, $\alpha = 0.36$ according to the results of this study. This gives the following equation for the salinity of outflowing water from the Baltic Sea

$$S_{out} = \alpha S_0 \frac{Q_{in}}{Q_f + \alpha Q_{in}} . \tag{23}$$

Assuming that $\alpha$ and $S_0$ are independent of $Q_f$, the derivative with respect to $S_{out}$ can be written as

$$\frac{\partial S_{out}}{\partial Q_f} = \alpha S_0 \frac{-Q_{in} + \beta Q_f}{(Q_f + \alpha Q_{in})^2} \tag{24}$$




where $\beta = \frac{\partial Q_{in}}{\partial Q_f} = -0.46$ according to the results above. With $S_0 = 33.5$ psu, $Q_{in} = 13500$ m³/s and $Q_f = 15700$ m³/s this gives

$\partial S_{out}/\partial Q_f$ = - 5.91 10⁻⁴ psu s m⁻³ or 1.05% decrease in outflow salinity for 1% increase in $Q_f$. The finite difference derivative

based on mean outflow salinities and net fresh water input of the RP+ and RP- experiments is $\Delta S_{out}/\Delta Q_f$ = - 5.23 10⁻⁴ psu s m⁻³ or 0.93% decrease in outflow salinity for 1% increase in $Q_f$. The two estimates are similar enough to support the assumptions

behind the simplified steady state expressions, keeping in mind that $\alpha$ is not perfectly constant, that the runs are not totally in steady state, and that diffusive transports across the boundaries are not considered.

With $\beta = 0$, i.e. with no decrease in inflow for increasing freshwater input, (24) gives $\partial S_{out}/\partial Q_f$ = - 3.85 10⁻⁴ psu s m⁻³ which is 65% of the value with $\beta$ = -0.46. This means that 35% of the sensitivity of Baltic Sea salinity to freshwater input can be attributed to decreased inflows alone.

If we instead assume that the inflow salinity at the sill transect is constant (i.e. not governed by eq. 22), the outflow salinity can be found directly from the Knudsen relations as

$$S_{out} = \frac{S_{in} Q_{in}}{Q_f + Q_{in}} . \tag{25}$$

with the partial derivative with respect to $Q_f$ being

$$\frac{\partial S_{out}}{\partial Q_f} = S_{in} \frac{-Q_{in} + \beta Q_f}{(Q_f + Q_{in})^2} . \tag{26}$$

With a constant inflow salinity of 16 psu corresponding to the steady state reference run, this gives $\partial S_{out}/\partial Q_f$ = - 3.89 10⁻⁴ psu s m⁻³, which is 66% of the full sensitivity. This means that 34% of the sensitivity can be attributed to decreased inflow salinity alone, i.e. by recirculation of outflow water in the Kattegat and Belt Sea region.

The sensitivity with both constant inflow and constant inflow salinity, i.e. caused by dilution of the Baltic Sea alone, can be determined from (26) with $\beta = 0$ and is 43% of the total sensitivity. The combined influence of reduced inflows and reduced

salinity due to recirculation in the entrance region therefore is 57% of the total sensitivity. These simplified estimates based on results from the full sensitivity study cannot be expected to give the exact picture, but they do show that dilution inside the Baltic Sea, inflow changes, and inflow salinity changes due to recirculation in the entrance region all contribute significantly to the Baltic Sea sensitivity to fresh water forcing.

A sensitivity of about 1% salinity decrease for a 1% inflow increase is in the same order of magnitude but in the low end of

earlier estimates (e.g Stigebrandt 1983, Stigebrandt and Gustafsson 2003, Meier and Kaufer 2003) which are generally in the range 1.15 - 1.5%. None of these studies have explained the sensitivity by the simple processes presented here.

The sensitivity of Baltic Sea outflow salinity to North Sea salinity can easily be determined from (23) as

$$\frac{\partial S_{out}}{\partial S_0} = \alpha \frac{Q_{in}}{Q_f + \alpha Q_{in}} , \tag{27}$$

which gives 0.24. This is smaller than 0.37 which is found from the finite difference gradient of outflow salinities based on

the SB+ and SB- runs, which may again have to do with the lack of totally steady conditions. Note that according to this expression the relative change in the Baltic Sea is expected to be equal to the relative change in the North Sea. It is interesting

to note that Stigebrandt (1983) found a sensitivity of 0.3 with a model much simpler than our regional model but also much more complex than (23) and (25).

The validity of the presented results depends on the validity of the model. The model has shown a good representation of
historical variability of salinities in the Baltic sea in response to varying freshwater input and atmospheric forcing, which lends some confidence in the conclusions.

The next step is to look further into the sensitivity of Baltic sea salinities to sea level rise. With sea level rise, both the recirculation of water in the Kattegat and Belts sea region and the inflow strength will change. Both the fraction of Skagerrak water in the inflows and the inflows themselves will increase. It is uncertain to what degree the present model is able to describe
these changes, both with regard to the basic coarse resolution of the bathymetry and straits where small cross-sectional changes may not be well represented, and with respect to resolving important physical processes governing the changes in inflow strength and mixing (e.g. Arneborg 2016). Given the large importance of recirculation and mixing in the entrance region for Baltic sea salinity, it will be necessary to look further into how well the model and possibly higher- resolution versions of the model represent water mass transformation processes in this region.

**Data availability**

Data will be made available on request.

**Author Contributions**

All authors conceptualized the study based on an idea of MH. Model set up was done by PP and model runs were performed by YL. All authors contributed to data analysis and visualization. LA led the manuscript writing with inputs from all authors.

**Competing Interests**

The authors declare that they have no conflict of interest.

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
