# Peer review of "Response of a semi-enclosed sea to perturbed freshwater and open ocean salinity forcing"

_EGUsphere, 2025_

## Referee Comment (RC1)

**Comments to: Response of a semi-enclosed sea to perturbed freshwater and open ocean salinity forcing**

**Suggestions for additional references**

Some studies dealing with the connection of precipitation / runoff, inflows, and sea level:

- H. Schinke, W. Matthäus, Continental Shelf Research 18 (1998) 67-97
- https://doi.org/10.1029/2023GL103853
- https://doi.org/10.1111/j.1600-0870.2006.00157.x
- https://www.tandfonline.com/doi/abs/10.1111/j.1600-0870.2007.00277.x

Water exchange between the North Sea and the Baltic Sea:

- https://doi.org/10.1016/j.ocemod.2020.101585
- Bertil Håkansson, Geophysica (2022), 57 (1), 3–22
- https://www.tandfonline.com/doi/abs/10.3402/tellusa.v48i2.12063
- https://doi.org/10.5194/os-21-913-2025
- https://doi.org/10.1007/s10236-024-01626-7

**Changes in the North Sea:**

• https://link.springer.com/book/10.1007/978-3-319-39745-0 (e.g., chapters 3.2.3 and 6.3 and references therein)

**Specific comments**

Line 17: Maybe a last sentence about the significance / implications of the results in the abstract?

Line 20: salinity is not everywhere below 12 psu in the Baltic Sea (although in most parts this is true); it should be mentioned that not the average low salinity but also the strong horizontal and vertical salinity gradients are challenging for the ecosystems

Lines 22-24: Additional information to that study could be useful. How strong were the salinity decrease and temperature increase in that study? Which other "anthropogenic" factors were they compared to?

Line 26: rather cite original studies (Meier et al. 2017; <a href="https://doi.org/10.1007/s00382-016-3333-y">https://doi.org/10.1007/s00382-016-3333-y</a> and maybe also Meier et al. 2021; <a href="https://doi.org/10.1038/s43247-021-00115-9">https://doi.org/10.1038/s43247-021-00115-9</a>) instead of the review Meier et al. 2022

line 37: Average / typical depths of the mixed layer / permanent halocline and the seasonal thermocline could be given

line 39: According to the reference Mohrholz 2018 (table 3 therein), about half of the salt transport into the Baltic Sea is baroclinic. Hence, the statement in the next sentence that the main transport is sustained by small barotropic inflows, should also be reconsidered.

Lines 54-55: How is the representation of inflows improved? Compared to predecessors of the model that was used or compared to other models?

Line 72: Are SB and RP really defined relative to their unperturbed states (i.e., as some kind of Delta SB, Delta RP)? For me, the Taylor expansion (specifically, the terms (SB – SB\_0), (RP – RP\_0) and so on) rather looks as if they are defined in an absolute way.

Line 75: Would you expect an interaction between the two forcing terms? In one way or both ways? It's not obvious that / how they should interact.

Lines 86ff: How did you choose h and k? How do they compare to, e.g., interannual variations of SB and RP? How large is the uncertainty they cause in the discrete differences in equations 3-7?

Line 91: The considered time span (1990-2017) is quite short given the pronounced multidecadal variability of the system. This might add some uncertainty to the results

Line 140: What is meant by "turbulent mixing of the inflowing water masses to the region"? How can a water mass be mixed to a region? Do you mean it's mixed to the water masses that are present in that region?

Lines 150ff: You say "The upper envelope ranges between the surface and 250 m and uses 43 levels". How does that match with terrain-following coordinates? Do you always have 43 levels if the water depth is less than 250 meters and 43+13 if it's more than 250 meters (which is rarely the case in the Baltic Sea)? Are the depth levels otherwise equidistant at a certain grid point? You wrote in the introduction that this model has an improved representation of inflows (see an earlier comment of mine). How does this selection of coordinates improve the representation of inflows? Is it better than, for example, adaptive vertical coordinates (https://doi.org/10.1016/j.ocemod.2011.04.007)?

Lines 162, 163: you write "we ran the model three cycles repeating the same 10 years period using atmospheric, runoff and open boundary forcing from 1961–1970 so that model dynamics reached near-equlibrium level". Earlier in lines 90, 91 you wrote "we use runs over the period 1961-2017, with t1 = 1990 and t2 = 2017, which gives a thirty-year long spin-up". Now, is the spin up period 1961-1990 or was the spin up run for 30 years in the three cycles and then the actual runs were started from 1961? Or do you have one general spin up period and then another one for the perturbations? Maybe, a small schematic could facilitate the understanding of the experimental design.

Line 183: What about the strong Major Baltic Inflow in January 1993?

Lines 199ff: I don't fully understand the choice of the additional experiments to test the validity of the polynomial. RP++SB++ makes sense but why isn't there any equivalent negative experiment RP--SB--? How did you choose experiments 8 and 9? And shouldn't the number of "validation" experiments be a bit larger than three (although that would be computationally demanding)?

Lines 213ff: It could be interesting to also analyze the surface mixed layer and deep-water layer in the Baltic Sea separately for the different experiments. But maybe that's beyond the scope of the study.

Line 229: Is it really clear from the figure that the response to the perturbations is linear in h? For me, the contours are too coarse to tell

Line 232: How can you compare the values if they have different units?

Lines 245, 246: You write that (SB\_0 + 2h, RP\_0 + 2.5k) is quite a large extrapolation. However, if I get it right, the range in figure 6 is even larger or not (at least, the salinity range is larger than that in the previous figure)?

Line 269: "the net outflow, seen as Q at low salinities" – isn't it particularly seen at s = 0?

Line 276, 277: "At the sill transect, the influence of fresh water input on inflows is seen to be larger for low saline inflows than for high-saline inflows." Is it because low saline inflows contain a larger portion of fresh water which makes the impact of changes in freshwater input larger? Or is there a different explanation?

Lines 281, 282: "Changing boundary salinities are mainly affecting the salinities of inflows and outflows at the northern Kattegat boundary, but do cause less changes to the inflows and outflows at the sill transect." This is really difficult to see in the figures which is why I would suggest to have separate panels / inlets with only the maxima of the curves and their dependencies (see figure comments).

Lines 284-286: Could there also be a small effect due to the fact that the transect across the sills is not closed? See discussion of figure 2 in Radtke et al. 2020

Line 288: How realistic is Gamma = 1?

Lines 300-310: Those results sound quite interesting. Do you get similar results directly from your simulations when comparing periods with low and high freshwater input (of course, there could be confounding factors like changing wind fields; but it would to some extent provide some validation)? Could you compute the overturning streamfunction or something like that in the Kattegat?

Lines 315ff: I have the impression that figure 12 needs some more detailed explanation and interpretation. Can you say a few words on why the maximum of the curve is shifted to the negative side while the tails of the distribution look relatively similar (i.e., all in all, there seems to be some skew involved)? What implications does this have for the actual in- and outflows? Do I see it right that for large inflows the blue and red curves overlap? What does that mean?

Lines 329ff: Could you make a rough calculation of how much more / less barotropic flow across the sills can be expected for a few centimeters of change in sea level gradient as shown in figure 13 (you might use equation 3 from Mohrholz 2018 as in <a href="https://doi.org/10.1029/2023GL103853">https://doi.org/10.1029/2023GL103853</a>)?

Lines 349, 350: "When increasing precipitation and runoff to the Baltic Sea, 54% of the increased net fresh water input were exported as increased outflows through the sill transect, whereas 46% resulted in decreased inflows." I'm not sure this is correct (or I get it wrong). Your figure 11 shows volume fluxes as far as I understand, not freshwater fluxes. Also, the decreased inflow is (to some extent) a result of the increased outflow due to the recirculation you show in the figure while your statement sounds as if they are independent of each other.

Lines 352-354: "With the large-amplitude fluctuations in in- and outflows taking place in the inflow region, such a more or less constant net change to the barotropic flows contributes almost equally to increased outflows and decreased inflows." Wouldn't those modifications also be there if the amplitude of the fluctuations between in- and outflows was smaller? Or what's the message here?

Line 357: Shouldn't it be the "net outflow of volume"?

Line 363: How do you get from equation 22 to 23? I suppose you are employing the Knudsen relations?

Line 366: Where do you take  $S_0 = 33.5$  psu from? If I search the document for "33.5", I don't find it anywhere else. Is it from the TEF analysis (same question for the 16 psu in line 380)?

Line 371: At which steps in the calculations are diffusive fluxes neglected? How important are they? I think they are mentioned for the first time here

Lines 383-388: How do your results compare to other estimates like those of Radtke et al. 2020 or Meier et al. 2023? In the same paragraph, how exactly do you differentiate between inflows and inflow salinities? With inflow, do you mean the inflow volume which is sensitive to changes in the freshwater forcing due to the change in sea level gradient that you described before? It's important to be precise here because the term "inflow" is often used for both the volume and the salt import.

Line 391: How did the other studies explain their results then?

Lines 405-406: You mention that the model (of course) cannot properly resolve the Danish straits. Could you briefly mention in the model description of the methods section how you modified the bathymetry in the Danish straits to make sure that transports are realistic?

**Technical comments**

Line 8: Repetition "study" (alternative: "sensitivity experiments")

Line 8: Comma after "in this study" (also check comparable sentence structures)

Line 12: "Baltic Sea"

Lines 16 and 17: Better "Baltic Sea water" than "Baltic water"

Line 40-43: The sentence starting with "Smaller barotropic inflows..." is very long

Line 42: Generic abbreviation for Major Baltic Inflows would be "MBI", not "BMI"

Line 55: "the sensitivity experiment" – shouldn't it be plural?

Line 96: The comma seems wrong

Line 103: "through a transect" or "through transects"

Lines 153, 154: Is there a verb missing in "and the lower envelope all depths below 250 m using 13 levels"?

Line 233: comma missing ("the interaction term, the cross derivative, ...)

Line 277: "low saline" vs "high-saline"

Line 345: "Total exchange flow analysis ...", "sills" instead of "sill"

Line 411: Please check the requirements of Ocean Science for making data available (https://www.ocean-science.net/policies/data\_policy.html). If I get it right, you should make at least the data that was used to produce the figures and calculations available in some publicly accessible repository

Inconsistent spelling of "freshwater" and "fresh water" (e.g., title vs. abstract)

Inconsistent use of present and past tense (e.g., "showed" in line 22 vs "show" in line 24)

Inconsistent use of psu and g/kg

It seems as if there are no Acknowledgements

**Comments to figures**

Figure 1: The nonlinear axis scaling might be pointed out in the figure caption. In the caption, it should be "orange lines" instead of "orange line". In addition, basins mentioned in the paper should be labelled in the map such that readers from other regions understand where, for instance, the "Gulf of Bothnia" is.

Figure 2: Over which period where the modeled profiles averaged?

Figure 3: Units are missing in the depths of the stations given in the titles of the panels. Are the stations BY15 and BY31 really only 150 m deep in your model (80 m would also be quite shallow for BY5)? How were "surface water" and "bottom water" defined? Did you correct for a possible seasonal sampling bias (as, for example, in Radtke et al. 2020)? Do you have an idea why modeled bottom salinities at station BY15 are quite off at the end of the period? It looks as if the strong MBI in 2014 / 2015 was not captured that well.

Figure 4: The curves look very smooth. Is it really annual means or were they additionally smoothed? In addition, most curves don't look as if they reach a steady state in the last decades. Wasn't this a prerequisite for the Taylor expansion? (you mention it later in lines 284ff.)

Figure 6: Resolution is too low (also check resolution of other figures; they are not as bad but don't seem to be sufficient either). Also, I'm not sure whether I fully understand how the figure is composed. Do I get it right that you compute the Taylor coefficients (eq 3-7 plus reference salinity), then plug them into equation 2 and then vary SB and RP in equation 2 to explore how the salinity changes? Then this should maybe be reflected in the labeling of the x- and y-axes by labeling them "SB – SB\_0" and "RP-RP\_0" or so. Finally, is there a reason for the diverging colorbar? And if so, why is it centered around 8? Wouldn't it make more sense to center it around the reference salinity?

Figure 7: Axis labels are very small (maybe also check the other figures). You might also add a 1:1 line to better see deviations from the perfect correspondence. Is it RP + 2k? Or  $RP_0 + 2.5k$ ? (and also  $SB_0 + 2h$ )

Figures 8 and 9: Salinity units at the x-axis missing. Maybe, there could be a separate panel / inlet showing only the maxima (i.e., the points where outflow changes to inflow) – could be interesting to see how the x- and y-values of the maxima depend on the perturbation factors. What's the

resolution of your salt axis (is it large enough to properly resolve differences in s between the maxima?)?

Figure 12: y-label missing. In addition, although it's mentioned in the text, the caption should mention that the figure refers to the sill transect.

---

## Referee Comment (RC2)

Review of manuscript "Response of a semi-enclosed sea to perturbed freshwater and open ocean salinity forcing" by Arneborg et al. 2025

This study uses high-resolution model sensitivity experiments to examine how the Baltic Sea's steady-state salinity responds to variations in freshwater forcing and salinity at the boundary of North Sea. From these experiments, the authors constructed a second-order polynomial that relates the basin-mean steady-state salinity to changes in freshwater forcing and boundary salinity. Results show that the Baltic Sea's response to freshwater forcing is large and non-linear, whereas its response to boundary salinity is more linear but less significant. The authors also analyze the impact of freshwater forcing and boundary salinity changes on the freshwater volume fluxes in and out of Baltic Sea driven by circulation changes.

Overall, the manuscript presents some important results, but it needs some revising and reorganizing before it can be considered for publication. The detailed comments are provided below:

- 1. The study examines the sensitivity of Baltic Sea salinity to net freshwater input into both the Baltic and the North Sea, as well as to variations in salinity at the North Sea boundary. However, it is not clear which specific boundary of the North Sea is used for prescribing or evaluating the boundary salinity. Could you clarify this, and indicate the location on Figure 1? Additionally, it is very likely that changes in the net freshwater forcing would influence the boundary salinity in the North Sea itself. If so, how do you separate the individual impacts of the freshwater forcing and boundary salinity within your analysis?
- 2. The figure captions are vaguely written. For example, in Fig.2, it is not clear during which period the observed salinity profiles are compared with the hindcast profiles? Are these averaged over a period?
- 3. Figure 2: I suggest plotting the observed salinity profiles in thin lines rather than dots. The multiple grey dots at each depth level
- 4. Figure 3: Please report quantitative statistics for these comparisons. How do the mean and standard deviation of salinity differ between the observations and the hindcast experiments? It would be helpful to list metrics such as bias, root-mean-square error (RMSE), and correlation coefficients.

From the figure, it appears that the bias in the hindcast salinity increases with depth. To better illustrate this, you could include a vertical profile of correlation coefficients (with statistical significance indicated) to show how model–observation agreement varies with depth.

- 5. Figure 4 needs more explanation. Note that the salinity trends are different RP- and RP-SB- runs as compared to the CTL and all other runs. Can this be explained? When you say yearly salinities, do you mean the mean salinity values averaged over each year? The caption is not clear.
- 6. Figure 5: How do these salinity anomaly maps for different runs compare with the CTL run?
- 7. Figure 6 caption is too vague. Specify the units.
- 8. Figures 8 and 9: Over what period have these volume and salt fluxes been integrated? Is there no inflow of salt into the Baltic sea, given all positive values of F in Fig. 9?
- 9. Figure 11: In the caption, you need to mention that these are the differences in fluxes between the RP+ and RP- experiments.
- 10. I recommend moving the equations and the calculations estimating percentage changes in salinity due to variations in inflows, outflows, and related terms from the Summary section to the Results section. The final section should focus primarily on synthesizing and highlighting the main findings.
- 11. The manuscript would be strengthened if the authors placed their results in better context with previous studies. In the Introduction, the authors state that their work differs from earlier studies because the model includes the North Sea domain. Could the authors elaborate on what difference this inclusion makes for the analysis and interpretation of salinity variability in the Baltic Sea? How do the results compare with those from previous studies that did not include the North Sea?